# Care Providers’ and Parents’ Experiences with Implementing the Conversational Health Literacy Assessment Tool (CHAT)-Maternity-Care in the Netherlands: A Mixed Methods Study

**DOI:** 10.3390/healthcare13101173

**Published:** 2025-05-17

**Authors:** Evi M. E. Vlassak, Judit K. J. Keulen, Elina Miteniece, Rianneke de Ritter, Marijke J. C. Hendrix, Marianne J. Nieuwenhuijze

**Affiliations:** 1Research Centre for Midwifery Science, Zuyd University of Applied Sciences, Universiteitssingel 60, 6229 ER Maastricht, The Netherlands; 2Care and Public Health Research Institute (CAPHRI), Maastricht University, P.O. Box 616, 6200 MD Maastricht, The Netherlands

**Keywords:** health literacy, midwifery, health communication, maternal child nursing, patient-centered care, women’s health

## Abstract

**Background/Objectives:** Care providers’ understanding of patients’ health literacy is crucial to tailoring care and reducing health inequalities. This study explores the experiences, facilitators, and barriers encountered by maternity care providers when implementing CHAT-maternity-care: a conversational tool that supports care providers in estimating (expectant) parents’ health literacy. As a secondary objective, the study also examines the experiences of (expectant) parents. **Methods:** Maternity care providers used CHAT-maternity-care after finalizing an e-learning. Implementation was evaluated among maternity care providers with a questionnaire and in-depth focus group meetings and among (expectant) parents with semi-structured interviews. **Results:** Providers experienced that using CHAT-maternity-care enhanced their health literacy insight, improved health literacy awareness, and fostered easier, more comprehensive and structured estimation of parents’ health literacy. Key facilitators for implementing CHAT-maternity-care as perceived by providers were the perceived value of health literacy insights; the tool’s relevance, user-friendliness, and familiarity; and social factors. The main barriers were time constraints, the tool’s novelty, and social factors. (Expectant) parents were positive and open to having conversations based on CHAT-maternity-care. Questions based on CHAT-maternity-care were perceived as beneficial by parents in uncovering previously unaddressed concerns. **Conclusions:** CHAT-maternity-care is mostly well received and assessed as helpful to improving health literacy insights. The findings underscore the importance of education, peer support, and organizational alignment for broader adoption and implementation of CHAT-maternity-care.

## 1. Introduction

Health literacy (HL) is a key determinant of health inequalities [1]. HL refers to the cognitive and social skills that determine an individual’s motivation and ability to access, understand, and use health-related information to promote and maintain good health [1]. Limited health literacy (LHL), the limitation of these skills, affects a significant proportion of the population, with estimates varying from 25% to 72% across different countries [2]. It therefore poses a significant challenge to public health and healthcare systems. LHL is associated with adverse health outcomes, increased healthcare costs, reduced patient satisfaction, and barriers in effective communication with healthcare providers [3,4,5,6,7,8,9].

HL can be divided into four categories: functional, communicative, critical, and organizational HL [10]. The first three concern HL skills of the user of care. The fourth, organizational HL, refers to the role of services, organizations, and systems in providing accessible health information that aligns with varying levels of individual HL [11]. Improving organizational HL can help meet patients’ individual needs.

Healthcare organizations and providers should address LHL to ensure equitable access to appropriate care [12]. To optimally tailor care, healthcare providers need to estimate their patients’ HL. However, not all providers are consciously aware of the concept of HL [13]. Even if they are, they often struggle with estimating HL and do not systematically recognize individual HL needs [13,14,15,16,17]. This issue is also prevalent in maternity care [18], indicating that there is potential for improvement in recognizing and addressing HL of (expectant) parents among maternity care providers.

In collaboration with maternity care providers, (expectant) parents, and HL experts, a supportive conversational tool was developed to explore and estimate (expectant) parents’ HL within regular maternity care: CHAT-maternity-care [19]. CHAT-maternity-care is based on the validated Health Literacy Questionnaire [20] and the original Conversational Health Literacy Assessment Tool (CHAT) [21]. CHAT-maternity-care is a practical tool for maternity care providers, offering examples of questions and observations in four domains to guide estimations of (expectant) parents’ HL (Table 1). Domain 1 covers communicative HL and relates to parents’ feelings of being understood and supported by care providers, their ability to actively engage with care providers, and their ability to navigate the healthcare system. Domain 2 covers communicative HL and relates to parents’ feelings of being understood and supported by their social network when faced with health-related issues. Domain 3 covers functional and critical HL and relates to parents’ skills to find, understand, and appraise health information to manage their health. Domain 4 covers critical HL and relates to parents’ current health behavior and health promotion to improve their health. By addressing these four domains, providers estimate the functional, communicative, and critical HL of (expectant) parents, enabling them to tailor care to (expectant) parents’ HL.

CHAT-maternity-care was developed collaboratively with stakeholders to ensure it is culturally relevant, logistically sound, and well supported within the field [22]. Its development involved application, evaluation, and adaptation through iterative pilot rounds, with positive feedback on the final version [19]. Despite these encouraging evaluations, large-scale implementation remains essential for meaningful impact [23]. Large-scale implementation requires strategies tailored to both innovation and context-specific facilitators and barriers [24].

The final version of CHAT-maternity-care has not yet been broadly implemented, and its facilitators and barriers remain unevaluated. Jensen et al. [25] qualitatively assessed the feasibility of implementing the original CHAT with Danish rehabilitation providers and suggested that the tool is promising for assessing HL needs and raising providers’ awareness. They recommended a structured implementation program, including an introduction to HL and guidance on acting upon CHAT results to overcome barriers such as difficulty contextualizing HL and lacking tools to address HL needs. While Jensen et al. [25] identified characteristics of the adopting person as potential barriers, implementation research highlights additional determinants that can either hinder or facilitate implementation: characteristics of the innovation, socio-political context, and organization [26]. Since midwifery differs from rehabilitation services, CHAT-maternity-care is distinct from the original CHAT, and implementation research highlights additional determinants, different implementation facilitators and barriers may emerge. Therefore, the research question of this study is as follows: What are the experiences of maternity care providers with the use of CHAT-maternity-care, and what are the perceived facilitators and barriers to its implementation within the Dutch maternity care system? As a sub-question, we also explored the experiences of (expectant) parents with the implementation of CHAT-maternity-care.

## 2. Materials and Methods

In an observational mixed-methods study, we investigated maternity care providers’ experiences, perceived facilitators, and perceived barriers when implementing CHAT-maternity-care and (expectant) parents’ experiences with it.

### 2.1. Setting

During this study, CHAT-maternity-care was implemented in the Dutch midwife-led primary care setting by community midwives and maternity care assistants. Almost 90% of pregnant women in the Netherlands start care in midwife-led practices [27], and nearly all women receive postnatal care from both community midwives and maternity care assistants. In Box 1, the organization of maternity care in the Netherlands is explained [28,29,30].

Box 1Organization of maternity care in the NetherlandsThe Dutch maternity care system is based on a division between primary care provided in the community and secondary and tertiary care in hospitals [29]. Women’s care is based on the assessment of the individual risk of each woman. Women with a low-risk pregnancy are cared for by community midwives in midwife-led primary care, and have the option of birthing at home, in a birth center, or having a midwife-led hospital birth. Women at intermediate or high risk are referred to obstetrician-led secondary or tertiary care, where they are looked after by hospital-based midwives, nurses, and obstetricians [28]. Women with low-risk postnatal periods spend their postpartum period at home, where the maternity care assistant, together with the community midwife, provides care during the first eight days after birth.Maternity care providers in primary, secondary, and tertiary care within a region work together in maternity care collaborations. Maternity care collaborations have a central role in maternity care policy development and in stimulating collaboration [30].

### 2.2. Implementation

Based on the recommendations of Jensen et al. [25], our study incorporated an educational component of HL and provided materials to tailor care to the HL of (expectant) parents. We developed an e-learning to train maternity care providers in estimating (expectant) parents’ HL using CHAT-maternity-care and tailoring care accordingly. The e-learning provided (1) background information about HL and LHL, (2) information about CHAT-maternity-care, and (3) tips, tools, and existing materials to tailor care to (expectant) parents with LHL. After completing the e-learning, providers applied CHAT-maternity-care in practice for 6–8 weeks before data collection started.

### 2.3. Participants

This study primarily explored the research question from the perspective of maternity care providers but also incorporated perspectives of (expectant) parents, recognizing its importance.

#### 2.3.1. Maternity Care Providers

Community midwives and maternity care assistants were recruited for the e-learning, implementation, and evaluation. They were recruited individually through convenience sampling by sending emails to midwifery care practices and maternity care organizations in the south of the Netherlands, and by posting an advertisement on the Maastricht Midwifery Academy website and social media. Additionally, snowball sampling was used, where initial participants forwarded the invitation email to others who met the study criteria. Participants signed up via email.

#### 2.3.2. (Expectant) Parents

Eligible participants included Dutch- or English-speaking (expectant) parents who were pregnant or gave birth between March and November 2023. Hereafter, “parents” refers to both current and expectant parents. Participants were recruited through purposive sampling by midwifery students during their clinical placements in midwifery practices and randomly in a midwifery practice waiting room by the research team. Potential participants received verbal information about the research’s objective and relevance, practical details, and inclusion criteria.

### 2.4. Ethics

The ethics board of Maastricht University Medical Centre determined that this research is not subject to the Medical Research Involving Human Subjects Act (WMO) and does not require ethical approval (number 2022-3283). Participants gave written consent before participation, participated voluntarily and anonymously, and could withdraw anytime, without consequences for parents’ care. Data were securely stored on a server accessible only to the research team.

### 2.5. Data Collection Among Care Providers

In June and July 2023, data from maternity care providers were collected after they had used CHAT-maternity-care with a questionnaire, followed by in-depth focus group meetings.

#### 2.5.1. Questionnaire

The questionnaire was developed based on the RE-AIM framework [31,32]. This framework guides planning and evaluation of implementation of public health interventions. The questionnaire included 32 questions, both open-ended and closed. The first five covered background information (e.g., age, work experience), followed by questions based on the five components of the RE-AIM framework (Appendix B):**R**each: Two questions addressed whether CHAT-maternity-care reached the target group.**E**ffectiveness: Three questions explored CHAT-maternity-care’s anticipated impact.**A**doption and **I**mplementation: Seventeen questions examined perceived facilitators of and barriers to adoption and implementation. To ensure that no aspect was overlooked, response options of two closed questions regarding barriers and facilitators incorporated 26 of the 29 validated determinants for the implementation of innovations [33]. Three determinants were excluded after discussion in the research team, as they were not applicable to our implementation context.**M**aintenance: Five questions addressed future use and recommendations for broader implementation.

#### 2.5.2. Focus Group Meetings

Focus group meetings aimed to qualitatively deepen the questionnaire results. The focus group guide (Appendix B) addressed notable experiences, facilitators, and barriers in all five key outcomes of the RE-AIM framework that emerged from the questionnaire. Six online focus group meetings were conducted via Teams, each with five to eight maternity care providers. Two research team members conducted the focus group meetings (EV attended all, alternately assisted by RR, JK, EM). The meetings were recorded and transcribed verbatim.

### 2.6. Data Collection Among Parents

Data from parents were collected in September and October 2023. In individual semi-structured interviews, parents were asked whether each CHAT-maternity-care domain had been discussed with their care provider. If they responded affirmatively, they were invited to share their thoughts on the topic; if not, the interview proceeded to the next domain. Subsequently, they were presented with the questions from CHAT-maternity-care (Table 1) to evaluate their comfort level discussing each domain. EV conducted the interviews, either online via Teams or in person at the midwifery practice. The interviews were recorded and transcribed verbatim.

### 2.7. Data Analysis

Data from closed questions were analyzed using descriptive statistics in SPSS 29. Open-ended questions were analyzed using inductive content analysis [34]. Coding was conducted by EV on a question-by-question basis and reviewed by EM. Focus group transcripts were deductively coded based on the five components of the RE-AIM framework [31,32,34]. Subthemes within these components were identified and categorized inductively through an iterative process by EV. Once no new subthemes emerged from the focus group transcripts (i.e., thematic saturation was reached), all transcripts were re-coded to ensure consistency and to confirm that no relevant data had been overlooked. Interview transcripts were deductively coded by EV according to the domains of CHAT-maternity-care [34]. Coding of the focus group and interview transcripts was conducted separately in NVivo 11 and peer reviewed by JK and EM. Additionally, the coding of the focus group data was checked independently by four midwifery students. Disagreements were resolved within the research team. Quotations were translated into English.

### 2.8. Rigor and Reflectivity

Methodological rigor was ensured using several strategies [35]. Data collection involved different methods, ensuring methodological triangulation. At the end of each focus group meeting or interview, the interviewer summarized the discussion for participant confirmation. Investigator triangulation was ensured by having multiple researchers analyze the data.

We used the concept of information power to ensure an adequate sample size [36]. A focused aim and specified participant target group enhanced data richness. Structured focus group meetings and interviews, guided by a theoretical framework and an experienced research team with a background in maternity care and/or health sciences, ensured meaningful exchanges. The analysis, which targeted implementation experiences, facilitators, and barriers, enhanced the data’s value. These considerations ensured sufficient, rich data collection while aligning with the concept of data saturation.

## 3. Results

The experiences with, perceived facilitators of, and barriers to implementation of CHAT-maternity-care are described for maternity care providers (Table 2, Table 3 and Table 4) and for parents. For providers, outcomes for each RE-AIM item are outlined based on the closed-ended questions in the questionnaire and further explored using insights from the open-ended questions and findings from the focus groups.

### 3.1. Care Providers

The study involved 14 maternity care assistants (age 24–59, mean age 48.6, work experience 1–36 years) and 23 midwives (age 23–57, mean age 33.8, work experience 0.5–32 years). All 37 participants started implementation and completed the questionnaire. Three participants withdrew due to time constraints. In-depth focus group meetings included 13 maternity care assistants and 21 midwives. The focus group meetings lasted 39–49 min.

#### 3.1.1. Reach

Most participants did not apply CHAT-maternity-care to every parent (Table 2). They mentioned that they made preliminary assessments and then targeted its use (Table 4). Some based their assessments on specific observations related to the four domains of CHAT-maternity-care, while others relied on intuition and past experiences.


*“I think that I mainly applied it to people whom I believed had LHL based on my experience from previous consultations.”*
Focus group-MCP1 (midwife)

Participants reported a greater tendency to use CHAT-maternity-care with parents with a lower educational level, young age, poor living conditions, or language barriers (Table 4). However, some participants were cautious, having seen highly educated, native-speaking parents show signs of LHL. They recognized that applying CHAT-maternity-care to all parents could lead to unexpected and insightful outcomes.


*“One of the most common mistakes is overestimating patients.”*
Focus group-MCP7 (midwife)

#### 3.1.2. Effectiveness

Participants mentioned CHAT-maternity-care was useful for discussing HL with parents. They mentioned that their estimation of parents’ HL improved when using CHAT-maternity-care (Table 2 and Table 4) and believed this could enhance the provision of appropriate care. Participants mentioned that CHAT-maternity-care facilitated more structured, comprehensive conversations and that its four domains with example questions and observations helped them to estimate parents’ HL faster and easier (Table 4).


*“People with LHL can often hide this well if you don’t ask the right questions. Through CHAT-maternity-care, this [the HL of the people] becomes clearer to me. In this way, we can recognize more and provide more specific and client-centered care.”*
Questionnaire (midwife)

Participants reported increased HL awareness due to using CHAT-maternity-care, prompting them to reflect on their previous behavior concerning parents’ HL (Table 4).

The answers in the questionnaire on perceived patient satisfaction differed: 25 of the 37 responders answered neutrally, and 11 of the 37 responders answered positively (Table 2). In the focus groups, participants explained that parents seemed to perceive it as a routine part of the process.


*“What I noticed is that it naturally came up in the conversation [the questions of CHAT-maternity-care] and they [the parents] thought it was part of it, that it was supposed to be that way. […]. They [the parents] just found it self-evident and were fine with talking about it.”*
Focus group-MCP29 (maternity care assistant)

#### 3.1.3. Adoption

Most participants had a positive attitude towards implementing CHAT-maternity-care in practice after getting acquainted with it (Table 2). Most participants mentioned that they found it important for estimating parents’ HL (Table 2, Table 3 and Table 4), valued the insights into HL from using CHAT-maternity-care, and believed estimating HL to be part of their job (Table 3 and Table 4).


*“I absolutely think it [using CHAT-maternity-care] is relevant because it provides much more insight into the HL of a particular parent.”*
Focus group-MCP33 (midwife)

Other facilitators mentioned for adoption of CHAT-maternity-care were the well-grounded content (Table 3) and providers’ willingness to learn something new and thereby continue their professional development (Table 4). A barrier to adoption was resistance to standardized approaches (Table 4).


*“It gives the feeling that even more protocols need to be followed and more lists need to be filled out. […] As a result, people may develop a negative attitude towards it.”*
Focus group-MCP 9 (midwife)

Opinions on the suitability of CHAT-maternity-care for the population varied among providers: Of the 37 responders, 20 answered positively, while 14 provided a neutral response (Table 2). In the focus groups, some participants mentioned that they cared for many individuals with LHL and therefore found CHAT-maternity-care particularly suitable. Other participants mentioned that CHAT-maternity-care was suitable for everyone, as many individuals with LHL tend to be overlooked.

#### 3.1.4. Implementation

The use of the CHAT-maternity-care tool varied: Some participants discussed all four domains within a single consultation; others discussed only one or a few domains in each interaction (Table 2). Some participants used a hard copy of the CHAT-maternity-care during the conversation as a memory aid, while others memorized the domains with example questions and observations. Participants valued the tool’s flexibility and convenience for tailoring to individual needs and specific situations (Table 4).

Furthermore, in both the questionnaire and the focus groups, participants identified ease of application, task alignment, content familiarity, and accessibility as factors facilitating implementation of CHAT-maternity-care (Table 2, Table 3 and Table 4). Other facilitators mentioned were the provision of clear instructions (Table 3 and Table 4) and colleagues integrating CHAT-maternity care (Table 4).

The most frequently mentioned barrier to implementing CHAT-maternity-care, as identified in the questionnaire and focus groups, was a lack of sufficient time to use the tool in all consultations with all parents (Table 2, Table 3 and Table 4). In addition, some midwives said that lack of funding could hinder allocation of time to routinely apply CHAT-maternity-care (Table 4).


*“You don’t have endless time in your consultation, there’s simply no compensation for that. We can’t implement that.”*
Focus group-MCP30 (midwife)

Other barriers included language barriers between providers and parents and lack of CHAT-maternity-care integration by colleagues (Table 2, Table 3 and Table 4). Participants explained that they were unable to discuss insights gained from using CHAT-maternity-care with colleagues who did not use the tool. Participants also mentioned the lack of organizational agreement on using CHAT-maternity-care as a barrier (Table 3 and Table 4).

Some participants experienced challenges with uncooperative parents (Table 3 and Table 4), noting a lack of understanding why certain questions were asked. Additionally, some participants considered it taboo to ask certain example questions from CHAT-maternity-care, finding them too personal. However, other participants mentioned that conversations went smoothly, and they perceived that parents were satisfied with the conversations. These participants did not perceive a taboo around the questions. They emphasized that timing and how the question was asked were key factors, noting that it helped when the question aligned naturally with the conversation, when the provider felt comfortable, and when the provider had internalized the questions.


*“I do recognize that it really depends on how you present things. […] Sometimes you have to respond to something, but sometimes asking certain questions can make people feel uncomfortable. […] You have to be very careful with that.”*
Focus group-MCP12 (maternity care assistant)

Since CHAT-maternity-care was new and the implementation period short, some participants found the application challenging. Those who used CHAT-maternity-care more frequently mentioned that they had internalized it, which facilitated its application. Participants who had not used the tool frequently shared this view and believed that frequent use would foster internalization and ease of use (Table 4).

#### 3.1.5. Maintenance

Most participants intended to continue using CHAT-maternity-care (or parts of it) and would recommend it to colleagues (Table 2). To further integrate CHAT-maternity-care into the Dutch maternity care system, participants proposed the following recommendations for implementation: (1) Incorporate it into curricula of midwifery and maternity care assistants’ education, (2) initiate training sessions within maternity care collaborations, (3) offer training programs through quality registers for midwives and maternity care assistants, (4) make CHAT-maternity-care available at a central location or send it by post to all maternity care providers, and (5) facilitate a standard note option for HL in patient records (Table 4).

### 3.2. Parents

The study included seven parents aged 19–38 (mean 29.4), with varying relationship statuses and educational levels (Table 5). Four were pregnant and three had given birth between March and November 2023. Interviews lasted 19–36 min.

#### Experiences of Parents

All parents expressed satisfaction with the maternity care they received. Interview findings showed that not all parents had previously discussed the four CHAT-maternity-care domains. Nonetheless, all parents agreed on the importance of discussing these domains with their provider and were open to such conversations.


*“They [the questions of CHAT-maternity-care] are clear. I can answer them easily. I don’t see any reason why I wouldn’t want to answer this.”*
Interview-P6

Regarding domains 1 (supportive relationship with care providers) and 3 (health information access and comprehension), parents mentioned that in their conversations with care providers, they mostly received advice rather than being asked questions. Some parents preferred this approach and felt comfortable with conversations focused primarily on receiving health information. Others said that being asked questions would have helped surface unaddressed concerns.


*“These are questions that can be asked, especially because there is sometimes an assumption that people might already know.”*
Interview-P1


*“I would want advice myself. I would not know what to do with this question [how to answer the question about who to contact with questions about pregnancy (domain 1)].”*
Interview-P2

Parents mentioned that social network support for discussing health-related topics (domain 2: supportive relationship within parents’ personal network) had not been previously discussed. Parents mentioned that they had no objection to talking about their social network support with their provider. Some mentioned that they would find it beneficial. However, others did not see the necessity of discussing health-related topics with anyone other than a healthcare provider.


*“Those are also important questions [Domain 2] […] I would have benefited from being asked those kinds of questions.”*
Interview-P1


*“No, that did not come up [questions about Domain 2], but I did not particularly feel the need for it either. However, I can imagine that if you have less social support around you, it would be good for the care provider to be aware of that.”*
Interview-P4

Parents said that they valued conversations with their maternity care provider about current health behavior and health promotion (domain 4), as they could highlight unrecognized issues or unhealthy handling of situations. The questions in domain 4 of CHAT-maternity-care were considered helpful, yet somewhat confrontational.


*“I also find these questions great because I think women might not be inclined to talk about things they might be ashamed of otherwise. […] I think it can be very important.”*
Interview-P6

## 4. Discussion

This study investigated experiences, perceived facilitators, and barriers when implementing CHAT-maternity-care. Providers experienced that using CHAT-maternity-care enhanced their HL insight, improved HL awareness, and fostered easier, more comprehensive and structured estimation of parents’ HL. Perceived facilitators by providers included the perceived value of gaining HL insights, alignment with professional roles and tasks, clear instructions, ease of use, accessibility, flexibility, familiarity, suitability for the population, and the tool’s well-grounded content. Perceived barriers included limited time and funding, perceived resistance from parents and providers, and language barriers. Peer adoption was perceived as a facilitator, while its absence and potentially linked lack of organizational agreements were perceived as barriers. The tool’s novelty was seen by some providers as a barrier, while others perceived it as a valuable opportunity for professional development. Parents were positive and open to having conversations based on CHAT-maternity-care. They perceived questions based on CHAT-maternity-care to be beneficial in uncovering previously unaddressed concerns.

In this study, within the components of reach and implementation of the RE-AIM framework, different maternity care providers indicated they used CHAT-maternity-care in various ways. Some used it selectively, while others applied it more comprehensively; some addressed all domains at once, while others focused on specific domains within a consultation; some used a hard copy, while others memorized the domains. This flexible use aligns with previous research on CHAT and CHAT-maternity-care, which emphasizes that providers can choose which domains and questions to use based on the parents’ context. It can be fully or partially integrated into existing assessments, depending on what is most useful and feasible in the specific care setting [19,21,25].

The providers’ experiences and perceived facilitators identified in the current study, within the components of effectiveness, adoption, and implementation of the RE-AIM framework, largely align with the findings of Jensen et al. [25] on the original CHAT implementation in rehabilitation services. The findings of our study indicate that CHAT-maternity-care might have the potential to effectively address key organizational HL barriers, such as limited HL awareness and the complexity of tools [37,38]. To support further implementation, it is essential to highlight providers’ perceived effectiveness and strengthen identified facilitators. This includes ensuring the tool’s accessibility through a centralized location and promoting its adaptability [23,39]. Additionally, developing and distributing a training program on HL and CHAT-maternity-care, such as an e-learning and interactive training session, into the bachelor’s midwifery curriculum and maternity care assistants’ education can be part of an implementation strategy [40]. Future research is necessary to validate the perceived effectiveness of CHAT-maternity-care in improving HL awareness and insight, providing evidence-based support for its wider implementation.

Both this study and the study by Jensen et al. [25] identified the influence of peers as either a barrier or a facilitator for implementation. This aligns with principles from behavioral science, which recognize social factors, such as others’ expectations and behaviors, as predisposing influences on behavior [41,42]. Therefore, it is recommended that providers implement the tool collectively with their peers, rather than individually. To facilitate this, implementation and associated education can be conducted within maternity care collaborations or healthcare organizations. These collaborations or organizations can establish agreements on its use, organize clinician implementation team meetings, ensure formal approval by management, and appoint an implementation champion to guide and support collective implementation [37,40,43,44].

Previous research shows that acceptance by service users (patients and clinicians) is critical for uptake [23]. In our study, some providers reported perceiving patient resistance during CHAT-maternity-care-based conversations as a barrier to implementation, while others did not experience resistance and observed that parents viewed the process as a routine part of care. This discrepancy in findings between the providers might stem from variations in providers’ comfort levels in posing CHAT-specific questions that could potentially alter the interaction dynamic. In our study, some providers reported hesitation, stemming from a perceived taboo around asking this type of question, while others did not perceive this taboo. Notably, all parents in our study expressed openness to discussing the various domains of CHAT-maternity-care. This aligns with the findings of a study on conversational psychosocial assessments that showed that while women’s acceptability was high, healthcare professionals reported discomfort when addressing particularly sensitive topics [45]. The taboo or hesitation experienced by some providers in our study might be related to cultural factors such as the protective approach and lightheartedness that midwives aim to achieve. Levy [46] described the concept of “protective gatekeeping and steering,” referring to midwives selectively withholding or sharing information to protect both themselves and their patients. Levy [46] highlighted that midwives often hold strong views about what was safe, potentially dangerous, or undesirable, which influence the way they guided their patients. Other previous Dutch studies found that midwives use small talk, minimizing language, and humor to foster and protect the bond with their patients during consultations [47,48]. For successful implementation of CHAT-maternity-care, it is essential to address providers’ resistance as a critical barrier. Institutional factors, such as limited time and insufficient competency development, may contribute to providers’ hesitancy in addressing sensitive topics, resulting in the avoidance of such discussions or the adoption of an instrumental approach rather than the intended conversational one [46]. Therefore, the absence of education in HL within professional education programs may contribute to the existence of a perceived taboo. Consequently, education represents a key strategy for effective implementation.

Other reasons for the discrepancy in perceived patient resistance by providers and the openness of having conversations based on CHAT-maternity-care indicated by parents might be incomplete data saturation with parents, or a mismatch between parents’ perceptions and their actual experiences. Post-implementation research is necessary to fully explore parental perspectives.

Time constraints and funding limitations were perceived as significant barriers to implementation. These barriers are consistent with broader research pointing to high workloads, restricted consultation time, and financial constraints as common barriers in maternity care [49,50,51,52]. With interactive training or prolonged use of CHAT-maternity-care, familiarity is likely to increase as it becomes internalized. This will facilitate deployment, reduce time consumption, and potentially save time by improving communication between providers and parents.

### 4.1. Limitations and Strengths

A strength of the study is the involvement of both care providers and parents, capturing different perspectives from both groups. Additionally, we strived to enhance the reliability and validity of the findings through multiple data sources and different perspectives of the members of the research team [35]. Using the RE-AIM framework [31,32] provided comprehensive insights into CHAT-maternity-care’s perceived facilitators and barriers. 

Potential selection bias exists, as providers who were more aware of or interested in HL might have been more inclined to participate. Based on Rogers’ diffusion of innovation theory, we suspect that the study participants were mostly innovators and early adopters, rather than the majority and laggards [53]. This might affect the tool’s perceived acceptance and usability. Still, the study provides valuable insights into implementation facilitators and barriers. Furthermore, evaluating CHAT-maternity-care’s impact on parents was challenging due to recruitment issues. Because of this, data saturation regarding the parents’ data might not have been achieved, potentially leaving some parents’ experiences being underrepresented. However, despite the small sample size, the included parents varied in individual and background characteristics. Therefore, the evaluation of CHAT-maternity-care was approached from various perspectives, providing a preliminary understanding of parents’ experiences.

Another limitation is that the questionnaire we used was not validated with the target population. However, it was developed based on established frameworks [31,32,33] and designed by researchers with expertise in health sciences and midwifery. Moreover, the questionnaire was not intended as a standalone measurement tool, but rather as an exploratory instrument to guide the focus groups. During these focus groups, the questionnaire findings were verified and further explored.

### 4.2. Recommendations

Future implementation and research should emphasize CHAT-maternity-care’s opportunities and address challenges through effective strategies. Education, peer support, and organizational alignment will be key in further promoting adoption and successful implementation. Further research is needed to validate the providers’ perceived effectiveness of the tool and to fully describe the perspective and experiences of parents. Beyond implementing CHAT-maternity-care, which facilitates the estimation of parents’ HL, future research should prioritize the next step: understanding the support providers need to effectively tailor their care to parents’ HL.

## 5. Conclusions

This study marks a relevant step in the implementation of CHAT-maternity-care by examining experiences, perceived facilitators, and barriers related to its use. The findings underscore the importance of education, peer support, and organizational alignment for broader adoption and implementation of CHAT-maternity-care within the maternity care context. Further research should focus on validating the perceived providers’ effectiveness of CHAT-maternity-care as well as further evaluation of parental experiences with it.

## Figures and Tables

**Table 1 healthcare-13-01173-t001:** CHAT-maternity-care [19].

Domain	Questions	Observations
1. Supportive relationship with care providers	Which care providers do you contact if you have a question about the pregnancy and the period thereafter? Do you know what questions to ask and which care provider to ask them to? Can you reach that care provider easily? How does it make you feel to talk to that person about the questions or concerns you have?	Are other care providers involved? How do parents respond to care providers who visit them during the postpartum period? Are the parents able to explain their problems/concerns well to you as a care provider?
2. Supportive relationship within parents’ personal network	With which people in your network (partner, family, friends, and neighbors) do you talk if you have questions about your pregnancy and the period thereafter? How does it make you feel to talk to that person/those persons? Do you feel understood by that person/those persons? Which person helps you best with health-related questions about you or your baby? How do they help you now? And how do you think they will help you in the future?	Is someone else coming along to appointments? Is this always the same person? After the baby is born, are there family, friends, and/or neighbors who can answer the parents’ health-related questions? Do the parents address each other’s health-related questions?
3. Health information access and comprehension	Did you search/are you searching for information about the pregnancy and the period thereafter? Where did you find/are you finding that information? Can you find this information easily or is it difficult? What do you think of this information? - Do you know what information you can trust and which not? - Is this information difficult or easy to understand? - Is it too much, too little, or just enough information? How do you compare different information (sources)?	What kind of questions do you receive from the parents? What information do the parents come to you with? What do parents do with the information they receive? Can they follow up on instructions? Are there signs that the parents have difficulties with writing or reading?
4. Current health behavior and health promotion	How do you take good care of yourself and your baby? What do you do on a daily or weekly basis to stay healthy? If you want to stay healthy during the period before and after the baby is born, what do you find easy and what difficult? Who or what helps you to live healthily during the pregnancy and the period thereafter? Who or what prevents this? What do you want to do to live healthily?	Are the parents actively involved in their health? Do the parents ask for help? Are the parents able to take steps to behave healthily?

**Table 2 healthcare-13-01173-t002:** Data from the questionnaire on the implementation of CHAT-maternity-care among maternity care providers (n = 37).

		Totally Disagree/Disagree n (%)	Neutral n (%)	Totally Agree/Agree n (%)	Yes n (%)	No n (%)
Reach	I applied CHAT-maternity-care to every patient.				1 (2.7)	36 (97.3)
Effectiveness	CHAT-maternity-care ensures that I gain better insight into the health literacy of (expectant) parents.	3 (8.1)	5 (13.5)	29 (78.4)		
	(Expectant) parents are generally satisfied when I use CHAT-maternity-care.	1 (2.7)	25 (67.6)	11 (29.7)		
Adoption	The insights into health literacy obtained using CHAT-maternity-care are valuable.	2 (5.4)	3 (8.1)	32 (86.5)		
	I find it important to gain insight into the health literacy skills of my patients.	2 (5.4)	0	35 (94.6)		
	I think CHAT-maternity-care is suitable for my patients.	3 (8.1)	14 (37.8)	20 (54.1)		
	It is part of my professional role to gain insight into the health literacy of (expectant) parents using CHAT-maternity-care.	2 (5.4)	5 (13.5)	30 (81.1)		
	After getting acquainted with CHAT-maternity-care, I had a positive attitude towards implementing it in practice.	3 (8.1)	6 (16.2)	28 (75.7)		
Implementation	I integrated CHAT-maternity-care into standard care.				10 (27.0)	27 (73.0)
	I used all four domains of CHAT-maternity-care in one conversation.				14 (37.8)	23 (62.2)
	CHAT-maternity-care aligns with my current way of working.	6 (16.2)	7 (18.9)	24 (64.9)		
	Using CHAT-maternity-care has personal advantages for me.	0	20 (54.1)	17 (45.9)		
	Using CHAT-maternity-care has personal disadvantages for me.	19 (51.4)	16 (43.2)	2 (5.4)		
Maintenance	I would recommend CHAT-maternity-care to colleagues.	3 (8.1)	11 (29.7)	23 (62.2)		
	I would continue using CHAT-maternity-care (or parts of it) in the future.				31 (83.8)	6 (16.2)

**Table 3 healthcare-13-01173-t003:** Perceived facilitators of and barriers to the implementation of CHAT-maternity-care mentioned by more than 25% of the maternity care providers (n = 37).

**Perceived Facilitators**	**Agrees n (%)**
Better estimation of health literacy	26 (70.3)
Easily accessible	19 (51.4)
Part of the professional role	19 (51.4)
Finding it important	17 (45.9)
Instructions are clear	16 (43.2)
Easy to use	14 (37.8)
Aligns with the current way of working	13 (35.1)
Sufficient knowledge, familiarity with the content	12 (32.4)
Well-grounded content	12 (32.4)
**Perceived Barriers**	
Insufficient time	25 (67.6)
Colleagues do not use it	13 (35.1)
No patient engagement, patient resistance	11 (29.7)
No agreement about the use within the organization	10 (27.0)

**Table 4 healthcare-13-01173-t004:** Experiences, perceived facilitators, and perceived barriers of implementing CHAT-maternity-care by maternity care providers, divided into the categories of the RE-AIM framework.

	Experiences	Perceived Facilitators	Perceived Barriers
Reach	Selectively applying based on a preliminary assessment considering the four domains ^1,3^ Greater tendency to use it with parents who have a lower educational level, a young age, or poor living conditions, or when language barriers were present ^3^		
Effectiveness	Better estimation of parents’ health literacy ^1,2,3^ Quickly and easily estimating the health literacy of parents ^3^ More comprehensive structured conversations ^1,3^ Increased awareness of health literacy ^1,3^ Reflection on their own previous behavior regarding the assessment and alignment with the health literacy of parents ^3^		
Adoption		Suitable for the population ^2,3^ Considering it important to estimate the health literacy of parents ^1,2,3^ Seeing it as part of their job responsibilities ^2,3^ Willingness to learn something new and thereby continue their professional development ^3^ The tool’s well-grounded content ^2^	Resistance to standardized approaches ^3^
Implementation		Flexibility of the tool ^3^ Clear instructions ^2^ Ease of application ^1,2,3^ Alignment with current tasks of maternity care providers ^2,3^ Adequate familiarity with the content ^2,3^ Accessibility ^2,3^ Colleagues also integrate CHAT-maternity-care ^3^	Lack of time ^1,2,3^ Colleagues did not integrate CHAT-maternity-care ^1,2,3^ No agreements about the use of CHAT-maternity-care within the organization ^2^ The existence of a language barrier ^1,3^ No funding ^3^ Perceived patient resistance ^1,2,3^ Hesitancy from providers to ask certain questions ^3^ The tool’s novelty ^1,3^
Maintenance	Recommendations for the future: Incorporate it into the curricula of midwifery education and maternity care assistants’ education. ^1,3^ Initiate training sessions within regional maternity care collaborations. ^1,3^ Offer training programs through the quality registers for midwives and maternity care assistants. ^1,3^ Make CHAT-maternity-care available at a central location or send it by postal service to all maternity care providers. ^1,3^ Facilitate a standard note option for health literacy in the patient records. ^3^		

^1^ = Main outcomes from open-ended questions in questionnaire, ^2^ = main outcomes from closed-ended questions in the questionnaire, ^3^ = main outcomes from focus group meetings.

**Table 5 healthcare-13-01173-t005:** Characteristics of (expectant) parents.

Participant	Age (Years)	Gravity, Parity, Abortion	Relationship Status	Educational Level
1	28	G1P1 after birth	Registered partnership	Higher professional education
2	26	G1P1 after birth	Living together	Secondary vocational education
3	31	G1P1 after birth	Living together	Higher professional education
4	33	G1P0 pregnant	Living together	University, master’s degree
5	19	G2P1 pregnant	Married	Pre-vocational secondary education
6	38	G5P3A1 pregnant	Living together	Secondary vocational education
7	31	G3P2 pregnant	Divorced	Primary education

## Data Availability

The data presented in this study are available on request from the corresponding author due to privacy restrictions.

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
