# Peer review of "Care Providers’ and Parents’ Experiences with Implementing the Conversational Health Literacy Assessment Tool (CHAT)-Maternity-Care in the Netherlands: A Mixed Methods Study"

_healthcare, 2025, doi:10.3390/healthcare13101173_

Round 1

Reviewer 1 Report

Comments and Suggestions for Authors

Review

Manuscript Title:
Care Providers’ and Parents’ Experiences with Implementing the Conversational Health Literacy Assessment Tool (CHAT)-maternity-care in the Netherlands: A Mixed Methods Study

General Comments
I appreciate the opportunity to review this manuscript, which addresses a highly relevant topic in the field of maternity care: the assessment of health literacy (HL) through a specific conversational tool (CHAT-maternity-care). The study presents a well-structured mixed methods design and is supported by the RE-AIM framework, which strengthens its methodological rigor.

However, I believe the manuscript would benefit from a series of improvements aimed at enhancing its clarity, analytical depth, and balance in the representation of participant voices. Below, I provide section-by-section observations intended to support the enhancement of the article:

There are some repetitions in the introduction, particularly concerning the definitions of health literacy. A revision is recommended to avoid redundancy.
Overall, the quality of the manuscript is good, and the topic is innovative and pertinent within the current context of person-centered care.

The study objective is adequately defined, though presented implicitly. It is recommended to reformulate it as a clear research question at the end of the introduction, which would contribute to a more analytically focused approach.
Relevant lines: 14–16 and 101–103
Original text: “This study aims to explore the experiences, facilitators, and barriers to implementing CHAT-maternity-care.”
Suggestion: Add an explicit formulation such as:

Methodology
The mixed methods design and the use of the RE-AIM framework are well justified. However, the sample of parents is small (n=7), which may affect the saturation of qualitative data and limit the generalizability of the findings.
Relevant lines: 360–362 (start of section 3.2)
Original text: “The study included seven parents aged between 19 and 38 years (mean 29.4)...”
Suggestion: Include a more robust reflection on this limitation in the Discussion section (lines 469–484), justifying the limited number of participants and analyzing how this may influence the interpretation of the findings.

Additionally, it is recommended to elaborate on the qualitative analysis process, particularly regarding the use of NVivo, the development of categories, and cross-validation strategies.
Relevant lines: 193–200
Original text: “Transcripts from the focus groups and interviews were analyzed separately using deductive coding in NVivo 11.”
Suggestion: Describe how the categories were constructed, whether double coding or peer review was applied, and how the reliability of the analysis was ensured.

It would also be helpful to clarify whether the inclusion of parents constituted a secondary objective of the study, as their analysis appears less developed compared to that of the care providers.

The results are well organized and clearly presented. However, there is some redundancy between the narrative and the tables. It is suggested to synthesize the narrative or transfer part of the information to the supplementary material.

Moreover, parents’ voices appear underrepresented. Including a greater number of direct quotes and diverse experiences would enrich the qualitative analysis and balance the representation of both groups.

The discussion is solid and includes appropriate comparisons with previous studies, such as Jensen et al. Facilitators, barriers, and the role of social and organizational factors are correctly identified.

Nonetheless, it is recommended to expand the analysis regarding professional resistance to using CHAT, particularly by exploring cultural and institutional factors such as the “gatekeeping” observed among some midwives. This dimension could provide better context for the reluctance identified.

The conclusion appropriately summarizes the main findings. Still, it is advisable to adopt a more forward-looking tone, including concrete practical implications and suggestions for future research.
Relevant lines: 494–504
Original text: “This study marks a relevant step...”
Suggestion: Add statements that emphasize the applicability of the model in other European or international contexts, and highlight that the tool is transferable with minimal cultural adaptations.

This manuscript holds strong potential and addresses a high-impact topic in clinical practice and public health. I recommend its revision to strengthen the analytical aspects, particularly in relation to the parent interviews, methodological description, and practical applications of the CHAT model.

I commend the research team for their work and encourage them to continue developing this valuable line of research.

Author Response

Dear Reviewer 1,

Point-by-point reply to reviewer1' comments:

Comment 1: There are some repetitions in the introduction, particularly concerning the definitions of health literacy. A revision is recommended to avoid redundancy.

Response 1: We agree there is some repetition in the introduction regarding the definition of health literacy. We have revised the introduction. The repeated explanation of the definitions has been removed (lines 50–53). Additionally, we have merged some sentences in the first paragraph to avoid unnecessary repetition (line 43).

Comment 2: Overall, the quality of the manuscript is good, and the topic is innovative and pertinent within the current context of person-centered care. The study objective is adequately defined, though presented implicitly. It is recommended to reformulate it as a clear research question at the end of the introduction, which would contribute to a more analytically focused approach.

Relevant lines: 14–16 and 101–103

Original text: “This study aims to explore the experiences, facilitators, and barriers to implementing CHAT-maternity-care.”

Suggestion: Add an explicit formulation such as:

Response 2: Thank you for this suggestion. In the abstract we formulated it more actively (line 14-19). New text:

“This study explores the experiences, facilitators and barriers encountered by maternity care providers when implementing CHAT-maternity-care: a conversational tool that supports care providers in estimating (expectant) parents’ health literacy. As a secondary objective, the study also examines the experiences of (expectant) parents”.

Also at the end of the introduction we have formulated it more explicitly as a research question (lines 102–110). New text: “Therefore, the research question of this study is: What are the experiences of maternity care providers with the use of CHAT-maternity-care, and what are the perceived facilitators and barriers to its implementation within the Dutch maternity care system? As a sub-question, we also explored the experiences of (expectant) parents with the implementation of CHAT-maternity-care.”

We also made adjustments in the first paragraph of the materials and methods section (line 112-114). New text: “In an observational mixed methods study we investigated maternity care providers’ experiences, perceived facilitators, and perceived barriers when implementing CHAT-maternity-care and (expectant) parents’ experiences with it.”

Comment 3: Methodology

The mixed methods design and the use of the RE-AIM framework are well justified. However, the sample of parents is small (n=7), which may affect the saturation of qualitative data and limit the generalizability of the findings.

Relevant lines: 360–362 (start of section 3.2)

Original text: “The study included seven parents aged between 19 and 38 years (mean 29.4)...”

Suggestion: Include a more robust reflection on this limitation in the Discussion section (lines 469–484), justifying the limited number of participants and analyzing how this may influence the interpretation of the findings.

Response 3: We agree that including only seven parents is a limitation, which we also reflected upon in our discussion. We have now expanded this reflection: “Furthermore, evaluating CHAT-maternity-care's impact on parents was challenging due to recruitment issues. Because of this, data saturation regarding the parents’ data might not have been achieved, potentially leaving some parents’ experiences underrepresented. However, despite the small sample size, the included parents varied in individual and background characteristics. Therefore, the evaluation of CHAT-maternity-care has been approached from various perspectives, providing a preliminary understanding of parents’ experiences.” (lines 547-553).

In the recommendations, we recommend to conduct further research to fully describe the perspective and experiences of parents with implementation of the CHAT-maternity-care (lines 564-566)

Comment 4: Additionally, it is recommended to elaborate on the qualitative analysis process, particularly

regarding the use of NVivo, the development of categories, and cross-validation strategies.

Relevant lines: 193–200

Original text: “Transcripts from the focus groups and interviews were analyzed separately using deductive coding in NVivo 11.”

Suggestion: Describe how the categories were constructed, whether double coding or peer review was applied, and how the reliability of the analysis was ensured.

Response 4: Thank you for this valuable feedback. We have expanded our description of the data analysis process accordingly (lines 209-225) New text: “2.7 Data analysis: Data from closed questions was analysed using descriptive statistics in SPSS 29. Open-ended questions were analysed using inductive content analysis [34]. Coding was conducted by EV on a question-by-question basis and reviewed by EM.

Focus group transcripts were deductively coded based on the five components of the RE-AIM framework [31,32,34]. Subthemes within these components were identified and categorized inductively through an iterative process by EV. Once no new subthemes emerged from the focus group transcripts (i.e., thematic saturation was reached), all transcripts were re-coded to ensure consistency and to confirm that no relevant data had been overlooked. Interview transcripts were deductively coded by EV according to the domains of CHAT-maternity-care [34]. Coding of the focus group and interview transcripts was conducted separately in NVivo 11 and reviewed by JK and EM.

Additionally, the coding of the focus group data was checked independently by four midwifery students. Disagreements were resolved within the research team. Quotes were translated into English.”

In ‘2.8 rigor and reflectivity’ we describe how the reliability of the analysis was ensured (lines 227-232).

Comment 5: It would also be helpful to clarify whether the inclusion of parents constituted a secondary objective of the study, as their analysis appears less developed compared to that of the care providers.

Response 5: Thank you for this helpful insight. Our main focus was on exploring the experiences, facilitators, and barriers encountered by maternity care providers when implementing CHAT-maternity-care, as they are the primary users of the tool. However, the perspective of parents could not be overlooked, as they are the ones to whom the questions are ultimately addressed. We already described this with other words in line 136-138. The focus groups with maternity care providers were also more extensive than the interviews with parents. We agree that exploring parents’ experiences was a secondary objective. We have now adjusted the description in the introduction accordingly:

“Therefore, the research question of this study is: What are the experiences of maternity care providers with the use of CHAT-maternity-care, and what are the perceived facilitators and barriers to its implementation within the Dutch maternity care system? As a sub-question, we also explored the experiences of (expectant) parents with the implementation of CHAT-maternity-care.” (lines 102-110).

Comment 6: The results are well organized and clearly presented. However, there is some redundancy between

the narrative and the tables. It is suggested to synthesize the narrative or transfer part of the

information to the supplementary material.

Response 6: Thank you for your feedback. However, we believe the tables complement the text, as they help to

clarify and contextualize the findings from the questionnaires alongside the narrative description.

We believe the narrative description is essential for presenting our results and providing a deeper understanding of them.

Comment 7: Moreover, parents’ voices appear underrepresented. Including a greater number of direct quotes

and diverse experiences would enrich the qualitative analysis and balance the representation of both groups.

Response 7: We acknowledge your remark that parents’ voices may appear underrepresented. Therefore, we

added more illustrative quotes from (expectant) parents to enrich the result section. (Lines 415-419 and 431-433).

Comment 8: The discussion is solid and includes appropriate comparisons with previous studies, such as Jensen et

Facilitators, barriers, and the role of social and organizational factors are correctly identified.

Nonetheless, it is recommended to expand the analysis regarding professional resistance to using CHAT, particularly by exploring cultural and institutional factors such as the “gatekeeping” observed

among some midwives. This dimension could provide better context for the reluctance identified.

Response 8: Thank you for this advise. In the discussion we elaborated more on the cultural factor “gatekeeping”

and institutional factors such as limited time and insufficient competency development (lines 506-521).

Comment 9: The conclusion appropriately summarizes the main findings. Still, it is advisable to adopt a more

forward-looking tone, including concrete practical implications and suggestions for future research.

Relevant lines: 494–504

Original text: “This study marks a relevant step...”

Suggestion: Add statements that emphasize the applicability of the model in other European or international contexts, and highlight that the tool is transferable with minimal cultural adaptations.

Response 9: In section ‘4.2 recommendations’, we provide suggestions for future research (lines 564-566).

We have revised the conclusion by removing the repetitive summary of results and adopting a more forward-looking tone. New text: “5. Conclusion: This study marks a relevant step in the implementation of CHAT-maternity-care by examining experiences, perceived facilitators, and barriers related to its use. The findings underscore the importance of education, peer support, and organizational alignment for broader adoption and implementation of CHAT-maternity-care within the maternity care context. Further research should focus on validating the perceived providers’ effectiveness of CHAT-maternity-care as well as further evaluation of the parental experiences with it.” (lines 569-581).

Comment 10: This manuscript holds strong potential and addresses a high-impact topic in clinical practice and public health. I recommend its revision to strengthen the analytical aspects, particularly in relation to the parent interviews, methodological description, and practical applications of the CHAT model.

I commend the research team for their work and encourage them to continue developing this valuable line of research.

Response 10: Thank you for your valuable comments. We highly appreciate the time and effort you have invested.

Reviewer 2 Report

Comments and Suggestions for Authors

The study provides a valuable example for health services research with the RE-AIM model.

It can be accepted with the answers to the following questions.
What criteria were used to decide on the sample size? What are Type I and Type II errors?
Seven parent data is relatively small. It is among the limitations of the study. It should be stated.

Author Response

Dear Reviewer 2,

Point-by-point reply to reviewer2' comments:

Comment 1: The study provides a valuable example for health services research with the RE-AIM model. It can be accepted with the answers to the following questions. What criteria were used to decide on the sample size?

Response 1: As this study is mixed methods with a larger qualitative and a smaller descriptive quantitative component, the emphasis lies on information power rather than sample size (as outlined in the section ‘2.8 rigor and reflectivity’, line 233-240: “We used the concept of information power to ensure an adequate sample size [36]. A focused aim and specified participant target group enhanced data richness. Structured focus group meetings and interviews, guided by a theoretical framework and an experienced research team with background in maternity care and/or health sciences, ensured meaningful exchanges. The analysis, which targeted implementation experiences, facilitators, and barriers, enhanced data’s value. These considerations ensured sufficient, rich data collection while aligning with the concept of data saturation.”

Additionally, this was an implementation study. The total number of maternity care providers who implemented the tool (n=37) matched the number who completed the questionnaire. As described in the results section, three providers withdrew, leaving 34 maternity care providers who participated in the focus groups (lines 252-254). Therefore, in the evaluation we included the entire population of eligible participants within the implementation context.

Comment 2: What are Type I and Type II errors?

Response 2: Since this is mostly qualitative rather than quantitative research, we did not formulate a hypothesis to either accept or reject. To minimize the risk of Type I errors (identifying patterns or themes that are not truly present) and Type II errors (overlooking real patterns or meanings) in the qualitative analysis, we employed strategies such as methodological and investigator triangulation. Additionally, we used member checking by summarizing the discussion at the end of each focus group or interview, allowing participants to confirm or clarify our interpretation. This has been described in the section ‘2.8 rigor and reflectivity’ (lines 228-232).

Comment 3: Seven parent data is relatively small. It is among the limitations of the study. It should be stated.

Response 3: Thank you for this feedback. Including only seven parents is a limitation of our study, which we reflected upon in our discussion. However, we have now expanded this section to more clearly eflect this limitation: “Furthermore, evaluating CHAT-maternity-care's impact on parents was challenging due to recruitment issues. Because of this, data saturation regarding the parents’ data might not have been achieved, potentially leaving some parents’ experiences underrepresented.

However, despite the small sample size, the included parents varied in individual and background characteristics. Therefore, the evaluation of CHAT-maternity-care has been approached from various perspectives, providing a preliminary understanding of parents’ experiences.” (Lines 547-553).

In the recommendations we recommend to conduct further research to fully describe the perspective and experiences of parents with implementation of the CHAT-maternity-care (lines 564-566).

Our main focus was on exploring the experiences, facilitators, and barriers encountered by maternity care providers when implementing CHAT-maternity-care, as they are the primary users of the tool. However, the perspective of parents could not be overlooked, as they are the ones to whom the questions are ultimately addressed. We already described this with other words in line 136-138. The focus groups with maternity care providers were also more extensive than the interviews with parents. Based on feedback of another reviewer we decided to describe the evaluation with parents was a secondary objective: “Therefore, the research question of this study is:

What are the experiences of maternity care providers with the use of CHAT-maternity-care, and what are the perceived facilitators and barriers to its implementation within the Dutch maternity care system? As a sub-question, we also explored the experiences of (expectant) parents with the implementation of CHAT-maternity-care.” (Lines 102-110).

Reviewer 3 Report

Comments and Suggestions for Authors

Thank you for the opportunity to review this study entitled: Care Providers' and Parents' Experiences with Implementing 2 the Conversational Health Literacy Assessment Tool 3 (CHAT)-maternity-care in the Netherlands: A Mixed Methods 4 Study.

The topic of the article is interesting mainly because of its possible applicability in practice. However, I consider it necessary that for its possible publication the authors should address several concerns that I detail below:

  • - In the methodology section, in the description of the instrument, no reference is made to any validation methodology. The authors designed their own questionnaire based on the RE-AIM framework, combining closed and open-ended questions, taking inspiration in part from validated determinants of innovation in health, but they did not carry out a formal validation process (neither cross-cultural adaptation, nor content validation, nor Cronbach's alpha type internal reliability). They incorporated 26 of the 29 validated determinants of innovation implementation but applied them “ad hoc” to the development of their questionnaire, without psychometric testing.
  • - Regarding the results, they seem confusing. They separate by dimensions (Reach, Effectiveness, Adoption, Implementation, Maintenance), but give very little explanation of the quantitative data: they mention percentages, but then only put a representative quote that exemplifies the trend, without going deeper. They do not analyze or discuss the nuances that might appear in open-ended responses or in the interviews.
  • - In the discussion they also do not explain the results well. They make general statements but do not analyze or elaborate on why and focus more on repeating generalities than on interpreting the findings in a critical and detailed manner.
  • In addition, they relate their findings to previous literature, but do not analyze in detail the differences or the particularities of their own context. In addition, they do not discuss in a timely manner the participants' quotations.
  • Among the limitations, they should put methodological limitations such as Suggesting future lines of research based on the results, not on generic ideas.

Author Response

Dear Reviewer 3,

Point-by-point reply to reviewer3' comments:

Comment 1: In the methodology section, in the description of the instrument, no reference is made to any validation methodology. The authors designed their own questionnaire based on the RE-AIM framework, combining closed and open-ended questions, taking inspiration in part from validated determinants of innovation in health, but they did not carry out a formal validation process (neither cross-cultural adaptation, nor content validation, nor Cronbach's alpha type internal reliability). They incorporated 26 of the 29 validated determinants of innovation implementation but applied them “ad hoc” to the development of their questionnaire, without psychometric testing.

Response 1: Thank you for highlighting the absence of a formal psychometric validation. We agree this is a limitation and have now made it explicit in the discussion (lines 554-559).

We developed a tailor-made instrument (appendix A) because existing validated questionnaires did not match our research scope, and complete validation of the questionnaire would not have been feasible within the project timeline and resources. The questionnaire was built on established literature (the RE-AIM framework [1,2] and MIDI [3] as described in ‘2.5.1. Questionnaire’ (lines 171-188)), and drafted collaboratively by researchers with midwifery and health-science expertise to improve its (face) validity. Importantly, the questionnaire was not used in isolation and served as an initial exploratory precursor to inform the focus groups. During the focus groups, the questionnaire findings were verified and further explored.

Methodological triangulation was achieved for many of the findings (Table 4). Only 3 out of 28 experiences/barriers/facilitators were mentioned exclusively in the closed-ended questions of the questionnaire. These three experiences/barriers/facilitators were determinants derived from the MIDI framework [3]. 25 out of 28 experiences/barriers/facilitators (also) emerged in the responses to the open-ended questions and/or in the focus groups.

  1. Holtrop JS, Rabin BA, Glasgow RE. Qualitative approaches to use of the RE-AIM framework: Rationale and methods. BMC Health Serv Res. , 18(1), 1–10.
  2. Glasgow RE, Vogt TM, Boles SM. Evaluating the public health impact of health promotion interventions: the RE-AIM framework. Am J Public Health. 2011, 89(9), 1322–7.
  3. Fleuren MAH, Paulussen TGWM, Dommelen P, Buuren S Van. Towards a measurement instrument for determinants of innovations. Int J Qual Health Care. 2014, 26(5), 501–10.

Comment 2: Regarding the results, they seem confusing. They separate by dimensions (Reach, Effectiveness, Adoption, Implementation, Maintenance), but give very little explanation of the quantitative data: they mention percentages, but then only put a representative quote that exemplifies the trend, without going deeper. They do not analyze or discuss the nuances that might appear in open-ended responses or in the interviews.

Response 2: Thank you for noting that the link between the quantitative and qualitative results could be clearer.

Because this is an exploratory study with a relatively small sample, we confined the quantitative analysis to descriptive statistics and illustrated the main patterns with representative quotations from the open-ended items and focus-group transcripts. Our aim was to highlight only the most notable questionnaire findings, presenting them with numerical context within the text. We believe that, for these selected examples, we provide an in-depth interpretation through the qualitative analysis, offering insight into how the questionnaire findings can be understood. We hope this explanation is clear and sufficient.

Example 1: “The answers in the questionnaire on perceived patient satisfaction differed: 25 of the 37 responders answered neutrally and 11 of the 37 responders answered positively (Table 2). In the focus groups, participants explained that parents seemed to perceive it as a routine part of the process.” (Lines 288-291).

Example 2: “Opinions on suitability of CHAT-maternity-care for the population varied among

providers: of the 37 responders, 20 answered positively, while 14 provided a neutral response (Table 2). In the focus groups, some participants mentioned that they cared for many individuals with LHL and therefore found CHAT-maternity-care particularly suitable. Other participants mentioned that CHAT-maternity-care was suitable for everyone, as many individuals with LHL tend to be overlooked.” (Lines 317-322).

Comment 3: In the discussion they also do not explain the results well. They make general statements but do not analyze or elaborate on why and focus more on repeating generalities than on interpreting the findings in a critical and detailed manner.

Response 3: Thank you for this feedback. In the discussion section, we already referred to the identified experiences, barriers, and facilitators. Sometimes we do this in a more general way (e.g. providers’ experiences and perceived facilitators, line 460). To clarify the relationship with the results of the study, we have now linked them explicitly to the components of the RE-AIM framework, which we had not sufficiently done previously (line 468-470; line 484; line 496; line 528).

In the discussion we choose to elaborate more on some of the specific barriers ( e.g. peer influence (line 483); time constraints and funding limitations (line 519); perceived patient resistance (line 486); hesitation (line 492)).

Comment 4: In addition, they relate their findings to previous literature, but do not analyze in detail the differences or the particularities of their own context. In addition, they do not discuss in a timely manner the participants' quotations.

Response 4: Thank you for this comment. We relate our findings to existing (Dutch) research in the discussion section of the paper. Several literature-based recommendations for implementation strategies echo our participants’ advice as discussed in the section ‘3.1.5. Maintenance’.

One of the differences identified in the results, and addressed in the discussion, is the perceived patient resistance. We have now expanded on this point in more detail (lines 493-526).

Additionally, we added a paragraph in the discussion section addressing the differences in reach and implementation.

New text: “In this study, within the components reach and implementation of the RE-AIM framework, different maternity care providers indicated they used CHAT-maternity-care in various ways. Some used it selectively, while others applied it more comprehensively; some addressed all domains at once, while others focused on specific domains within a consultation; some used a hard-copy, while others memorized the domains. This flexible use aligns with previous research on CHAT and CHATmaternity-care, which emphasizes that providers can choose which domains and questions to use based on the parents’ context. It can be fully or partially integrated into existing assessments, depending on what is most useful and feasible in the specific care setting [19,21,25]”. (Lines 459-467).

We trust this clarifies the relevance of our setting.

Comment 5: Among the limitations, they should put methodological limitations such as Suggesting future lines of research based on the results, not on generic ideas.

Response 5: We thank the reviewer for highlighting the need to sharpen the methodological limitations and to ensure that the suggested future lines of research should not be based on generic ideas. We have therefore expanded the methodological-limitations section to cover two study-specific issues: the small parent sample (n = 7) and the use of a non-validated questionnaire (Lines 547-559).

The recommendations for further research are mentioned earlier in the discussion and restated in the recommendations section. Based on our findings, the recommendations we make for future research include: investigating the providers’ perceived effectiveness of CHAT-maternity-care and further exploring the experiences and perspectives of parents (lines 564-566).

Although it is correct that the recommendation for research into understanding the support providers need to effectively tailor their care to parents’ health literacy (lines 566-568) is not directly based on the data of the current study, the recommendation is based on existing peer reviewed research from Jensen et al., which highlights the importance of this issue. In the current research we respond to this by providing support to healthcare providers, but we realized that there is currently limited knowledge about which materials professionals prefer to use when tailoring their care to parents with limited health literacy.

Round 2

Reviewer 3 Report

Comments and Suggestions for Authors

The authors have thoroughly addressed the  comments and have made the necessary revisions to improve the clarity and quality of the manuscript. Their responses demonstrate a clear understanding of the feedback provided.